# Mindfulness Teacher Trainees' Experiences (MTTE): An investigation of intense experiences in mindfulness-based interventions

**Erik Jönhagen**[1,2], **Tim Wood**[3], **Maria Niemi**[1], **Julieta Galante**[2,3] *

1 Department of Global Public Health, Karolinska Institutet, Stockholm, Sweden, 2 Department of Psychiatry, University of Cambridge, Cambridge, United Kingdom, 3 Contemplative Studies Centre, Melbourne School of Psychological Sciences, University of Melbourne, Melbourne, Australia

* mjg231@cam.ac.uk

## Abstract

With the increasing interest in mindfulness practices within clinical as well as non-clinical settings and the increasing body of research on the positive effects of mindfulness, concerns have been raised that mindfulness might also produce adverse effects including intense experiences and psychosis. The aim of this study was to investigate if intense experiences occur as a natural part of mindfulness practice, and if so to examine the characteristics of such experiences. We conducted a qualitative analysis based on fortnightly meditation reports from 13 mindfulness teacher trainees for 4 months. Intense experiences in meditation were frequently expressed in the reports of most of the practitioners and in some individuals these experiences were similar to psychotic-like experiences. This study presents suggestive evidence that mindfulness practices can produce intense experiences and that for some individuals these intense experiences may resemble psychotic-like experiences.

## 1. Introduction

### 1.1. Background

There is a continuously growing body of research regarding the topic of meditation in general and mindfulness-based interventions (MBIs) in particular in clinical and non-clinical contexts ranging from educational settings and corporations to various health-care settings world-wide [1]. MBIs are a collection of interventional approaches in which specific mindfulness meditation practices are used [2, 3]. The two most commonly used and researched types are mindfulness-based stress reduction (MBSR) [4, 5] and mindfulness based cognitive therapy (MBCT) [6]. The research on these interventions is extensive and show positive results across many domains including but not limited to stress-reduction, anxiety, depression, schizophrenia, pain and eating [7, 8].

In the scientific literature, the term "mindfulness" is used to describe quite different types of processes and it is subject to a variety of definitions [4, 9]. There is also an ongoing discussion regarding the scientific implications on the varied definitions of mindfulness across

**Data Availability Statement:** The data that informs the qualitative research presented in this manuscript cannot be made publicly available in order to protect the anonymity of the research

participants. However, excerpts of the transcripts are available within this paper, and further excerpts are available to researchers upon request to the Cambridge Psychology Research Ethics Committee (SBSEthics@admin.cam.ac.uk.

**Funding:** The authors EJ, TW, MN received no specific funding for this work. The author JG was funded by a United Kingdom National Institute for Health Research (NIHR, https://www.nihr.ac.uk/) Post-doctoral Fellowship (PDF-2017-10-018). All research at the Department of Psychiatry in the University of Cambridge is supported by the NIHR Cambridge Biomedical Research Centre (BRC-1215-20014) and NIHR Applied Research Centre. The funders had no role in the collection, analysis, or interpretation of the data or in the writing of the manuscript. The views expressed are those of the authors and not necessarily those of the NIHR or the Department of Health and Social Care.

**Competing interests:** The authors have declared that no competing interests exist.

domains [10]. In this paper we will use Kabat Zinn's definition as it is the most typical definition used in MBIs. Mindfulness is defined as "the non-judgmental acceptance and investigation of present experience, including body sensations, internal mental states, thoughts, emotions, impulses and memories, in order to reduce suffering or distress and to increase well-being" [4]. Mindfulness meditation is considered the main pedagogical faculty of MBIs [3]. MBIs encompass both pleasant and unpleasant experiences, with an emphasis on well-being and the inclusion of practices and psychoeducation aimed at addressing challenging emotions. Mindfulness meditation exercises employ a combination of open monitoring and focused attention methods. Focused attention is a way of focusing your mind on a specific object, for example the breath, and trying to stay with it throughout the meditation session. In meditation practices where open monitoring is used, the practitioner aims to have a wider area of focus and to pay attention to whatever may arise in the mind, such as thoughts, feelings, physical sensations etc. [7, 9]. Throughout MBIs approximately half the time is spent doing focused attention and half open monitoring [7, 9, 10].

Over the last few years a complementary focus has been on the potential adverse effects as well [11]. Modern mindfulness meditation practices are largely inspired by religious traditions, mainly Buddhism. In the Buddhist literature, it is considered obvious that regular meditation practice is not a linear highway into decreased stress and increased wellbeing. On the contrary, meditation is often described as a process in which both pleasant and unpleasant experiences will occur as a natural part of the practice [12]. Intense experiences in mindfulness meditation may be unpleasant in different ways, and even if they are pleasant, they could be unwanted, or leave the practitioner confused. In this paper intense experiences will be understood as a strong sensate or psychological experience that arises during mindfulness meditation that clearly differs from normal expectations of stress reduction. It is important to study the nature, context and predisposing factors of intense experiences to minimize the possibility of harm.

## 1.2. Mindfulness meditation and psychotic disorders

It has been suggested that the practice of meditation could induce psychosis in vulnerable individuals, as well as in people without history of psychiatric illness, although the evidence is scarce and to this date limited to case reports and some reviews [13–15]. To further complicate the picture, there is also some evidence that MBIs may be useful to patients with psychotic symptoms [16]. So it seems that mindfulness meditation could potentially be both a culprit and a cure regarding psychotic experiences. In 2017, a large-scale multi-cultural survey showed 25% of respondents reported unwanted effects of meditation practices related to mindfulness [17]. A systematic review of meditation studies between 1975 and 2019, the majority of which were focused on MBCT or MBSR, showed 55 studies out of 83 mentioning meditation related adverse events-of the total adverse events reported in these studies about 18% were psychotic or delusional symptoms [18]. Another study focused exclusively on variations of MBCT and found 6–14% of the sample experienced lasting bad effects, which included self disturbance, visual lights, and time space distortions [19]. A population-based survey of meditators in the United States found that 71% of them had some degree of lifetime exposure to mindfulness-based practices, and the rate of lasting bad effects of meditation (which implies a negative impact lasting over 1 month) was 10.4% [11]. There are also historical and religious claims that mindfulness meditation, if practiced diligently and consistently in and of itself, will produce intense experiences regardless of the practitioner's predisposition [20, 21]. Mixed-methods research through the Varieties of Contemplative Experience (VCE) project at Brown University has described a wide range of religious experiences which the practice of mindfulness meditation can occasion. The VCE study generated a taxonomy of phenomenological

experiences that informed the current study, and domains of experience which meditators reported as unusual or intense, including sensate, conative, and changes in sense of self, among others [20]. The potential of a correlation between mindfulness meditation and psychosis has also been suggested through case reports [22, 23]. However, it is important to note that in these cases, the individuals in question were involved in intensive mindfulness meditation retreats [24]. A distinction between psychotic illness and psychotic or psychotic-like experiences is necessary for the purposes of the study topic, as the prevalence of psychotic disorder is much rarer and more narrowly defined, than the prevalence of psychotic or psychotic-like experiences. In a recent meta-analysis, the pooled incidence of all psychotic disorders was found to be 26,6 per 100 000 person-years [25]. Psychotic symptoms, or psychotic experiences may well be experienced without an underlying psychotic disorder. In fact, some researchers suggest that psychotic experiences exist in the general population as a phenotype, as opposed to an everything-or-nothing kind of phenomena [26]. In a British study of self-reported psychotic symptoms in absence of a psychotic disorder, the prevalence was found to be 5.5% [27]. The incidence of subclinical psychotic experiences is substantially higher, in fact around 100 times higher, than the traditional incidence of psychotic disorders [28]. It is clear that mindfulness meditation as practiced in Buddhist traditions can elicit challenging and difficult experiences, some of which can be serious and long lasting [29]. It is therefore essential to investigate whether similar effects might arise in MBIs.

## 1.3. Mindfulness meditation and intense experiences

There has been some preliminary research conducted on mindfulness meditation effects other than those beneficial to mental health [11, 30–34]. Increasing concerns are being voiced that some people might have unexpected and unwanted experiences as a result from mindfulness practice, ranging from intense emotions to altered perception and psychosis, sometimes generating clinically relevant functional impairment [22, 26, 27]. Furthermore, it seems that subsets of people respond differently to the various aspects of mindfulness [5].

In recent years, the methodology of mindfulness trials has been criticized by researchers arguing that, in many trials, there are not sufficient methodological structures for detecting potential harm, unexpected or adverse effects [28–31, 35]. Therefore, conducting research in the field of mindfulness meditation poses methodological issues that need to be addressed [33, 36]. One large issue that should be addressed regarding MBIs is the lack of long-term longitudinal data [32]. Other important issues, described by Van Dam et al. [30] include an insufficient construct validity in measures of mindfulness, which results from the difficulty of finding a coherent definition of mindfulness, and potential adverse effects from practicing mindfulness—a topic that is of growing interest among researchers [31, 37, 38]. In addition, very few MBI trials actively measure adverse effects, and rely instead on passive monitoring, which can underestimate the actual frequency by more than 20-fold [30]. Baer further discusses the potential of a positivity bias due to the tendency of overrepresentation of positive results while negative findings are either not published or obscured by post-hoc subgroup analyses or creative re-interpretations [32]. These concerns are important to bear in mind when both reading and producing research on mindfulness meditation.

## 1.4 Aim

This study aims to begin to address some of the gaps and concerns regarding the potential risks connected to MBIs. The aim was to look for initial evidence of whether secularly committed mindfulness meditation practitioners have intense experiences, and if so, to investigate the nature and patterns of these experiences.

## 2. Methods and materials

### 2.1. Study design, population, and procedure

This was an online prospective longitudinal study following 13 secular mindfulness teacher trainees. We recruited online between July and September 2019 through mindfulness teacher networks in English-speaking countries around the world. Inclusion criteria were as follows: 1. Secular mindfulness-based programme (including but not limited to MBSR and MBCT) teacher-in-training, or a teacher who finished their training within the past year; 2. No regular (i.e. at least 20 minutes daily or near-daily) meditation practice until the year prior to starting teacher training; 3. No or little exposure to Buddhist teachings (e.g. have not attended Buddhist retreats). The reason we excluded participants with a Buddhist affiliation is because we wanted to understand the occurrence and appraisal of intense mindfulness meditation experiences among modern secular practitioners. Having a Buddhist worldview could have predisposed the participants to have specific spiritual experiences, or provided a specific spiritual framework to interpret any intense experience; 4. No other previous meditation teacher training (including Yoga teacher training); 5. Residing in the United States, United Kingdom Ireland, South Africa, Canada, Australia or New Zealand; 6. Over 18 years old.

After consenting the participants were asked to answer a baseline questionnaire asking about demographics and mental health. The participants also got to fill two questionnaires; one on mental wellbeing, using the Warwick Edinburgh Mental Wellbeing Scale 14 item version WEMWBS [39], range 14–70, where a higher score indicates a higher sense of wellbeing [40], and one on schizotypy, which is considered a marker of increased risk of psychotic episodes, using the Schizotypal Personality Questionnaire (SPQ), range 0–74 [41], where a high score indicates a high schizotypal personality.

Qualitative data on participant experiences with mindfulness meditation was then collected through prompting them to write about their experiences in short online questionnaires on a fortnightly basis. Online questionnaires are more anonymous than interviews, so the chances that people will share sensitive experiences may be greater.

The participants were specifically asked if they had any notable experiences since the last report, with suggestions of what those notable experiences could entail: "If you had particularly intense, puzzling, pleasant, disturbing or unexpected experiences since you last completed an MTTE survey, please describe them here. If there was nothing special to note, just say so.". This means that what participants told us in their reports was what they considered to be out of their ordinary experiences.

### 2.2. Data analysis

First, the sample was described in tabular format. The distributions of the SPQ and wellbeing scales were detailed using means and standard deviations, and compared with available normative data on each of the respective scales. An inductive qualitative content analysis on the data collected from our baseline questionnaire and the participants' fortnightly practice reports was conducted for our first aim [42]. Information power was used to define an adequate sample size [43]. The outcome corresponding to the first aim was a description, stemming from the qualitative data analysis, of participants' self-reported and described intense experiences in mindfulness meditation.

NVIVO software [44] was used for organizing the reports of the participants into nodes. A node can be understood as a description of some aspect of the participants' mindfulness meditation practice. For example, if the meditator reported energy surging through the body during the practice, the node would be "surging energy" and if more practitioners report the same

phenomena this would be added to the same node. What these nodes were conveying was not predefined, the nodes were chosen to as closely as possible convey the participants' described experiences. Several different nodes were constructed to contain information of the content of the mindfulness meditation practice. These nodes were then organized into a hierarchical node tree (see S1 Fig) with nodes organized into themes, themes organized into theme categories, and theme categories organized into three feeling tone arms (see 3.3).

The practice reports were expressed in free text and sometimes contained many different aspects of the participants' practice. One report could be coded in several different nodes, as different parts of the participants' experiences reflect different aspects. E.g., a participant could describe that they experienced a lot of relaxing sensations one day, and some unnerving experiences on another day. Thus, the relaxing sensations would be classified into one node, and the unnerving experiences into another. Sometimes mental and physical sensations might be difficult to discern as an emotion might be expressed as a physical sensation and vice versa. In such cases, the main characteristic, mental or somatic, was used for coding. On some occasions, there was also reason to code these experiences into both the mental and somatic theme categories.

### 2.3. Ethical considerations

The participants contributed data through an online diary. The data was anonymized for all but the researchers. Writing about difficult experiences may raise awareness of difficulties of various types and increase discomfort. Our main safety measure was a contact link to the research team provided in every questionnaire email invitation, and a reminder to contact us if they felt the need for more support. If any participant wrote to us expressing this need, we would put them in touch with an organization offering support specifically for people with mindfulness meditation-related difficulties. This study was granted ethical approval by the Cambridge Psychology Research Ethics Committee with permit number PRE.2019.043 (https://www.bio.cam.ac.uk/psyres, gathered 20191220, 14:38).

## 3. Results

### 3.1. Description of the sample

This section describes the participants who completed the baseline questionnaire and were eligible for participation in accordance with the inclusion criteria. In total 13 participants contributed with data every fortnight in their online mindfulness meditation reports. This added up to a total of 77 reports. As the participants entered the study at different times, there was some variance in the number of reports from each participant, ranging between 2–8 (mean 6.9). This means that at most participants have been contributing 16 weeks' and at least 4 weeks' worth of mindfulness meditation reports. Tables 1 and 2 show the demographic characteristics as well as the baseline variables of the participants. The age ranged between 27 years and 68, with a mean of 46.8 years, and most participants were women (69%). Most participants were from Australia (38.4%) and the UK (30.7%). All participants had at least some form of college degree. Many (46%) participants had a master's degree. With regard to religious views most people considered themselves spiritual but not religious.

Fig 1 shows the distribution of the wellbeing scores. The mean WEMWBS wellbeing score was 51.6 (standard deviation (SD) = 6.6) and similar to the mean score 50.7 (SD = 8.7) from the Scottish Health Population Survey (HEPS) which was used as the main population group when determining the validity of the wellbeing tool [27].

Fig 2 shows that 7 participants scored 10 or less on the SPQ with a mean SPQ score of 14.1 (SD = 14.2, median = 8). The mean SPQ score in our sample was slightly higher (14.1) than in a study of magical thinking and mindfulness conducted by Antonova et al. (11.62, SD = 6.38);

**Table 1. Demographic characteristics of the participants at baseline.**

| Age | | N (%) |
|---|---|---|
| | <40 | 3 (23) |
| | 40–59 | 7 (59) |
| | >59 | 3 (23) |
| | Prefer not to answer | 0 |
| **Gender** | | |
| | Female | 9 (69) |
| | Male | 3 (23) |
| | Other | 1 (7) |
| | Prefer not to answer | 0 |
| **Country of residence** | | |
| | Australia | 5 (38) |
| | Hong Kong | 1 (8) |
| | Scotland | 2 (15) |
| | United Kingdom | 4 (31) |
| | United States | 1 (8) |
| | Prefer not to answer | 0 |
| **Highest degree of education** | | |
| | Bachelor's degree | 4 (31) |
| | Master's degree | 6 (46) |
| | Some college/university (e.g. 'A' levels, tertiary colleges, specialist colleges) but no Bachelor's degree | 3 (23) |
| | Other | 0 |
| | Prefer not to answer | 0 |
| **Religion** | | |
| | Neither religious nor spiritual | 2 (15) |
| | Spiritual but not religious | 10 (77) |
| | Prefer not to answer | 1 (8) |

however, the non-meditator group in that study, which could be taken as a reference group, scored 14.50. In our sample, the SPQ scores were not normally distributed, so the median score may be a better reference score. Four participants scored very high compared to the group mean (14.1) and median (8) in the SPQ with 21, 27, 34 and 47 points. These participants did not deviate from the rest of the group in the wellbeing score (59, 47, and 53 points), with one exception (34 points) [45].

Regarding participants' mindfulness meditation experience and expectations at baseline, most participants practiced one session per day, and between 1–7 days per week. For most

**Table 2. Hierarchical structure of the themes underlying the qualitative analysis.**

| STRUCTURE OF THEMES | | | | | | | |
|---|---|---|---|---|---|---|---|
| **Feeling tones** | **Pleasant** | | **Unpleasant** | | **Neutral** | | |
| **Theme categories** | *Mental* | *Somatic* | *Mental* | *Somatic* | *Mental* | *Somatic* | *No notable experiences* |
| **Themes** | Affective | | Affective | | Perceptual | | |
| | Perceptual | | Perceptual | | Cognitive | | |
| | Sense of self | | Cognitive | | Awareness | | |
| | Magical thinking | | | | Deep/intense | | |

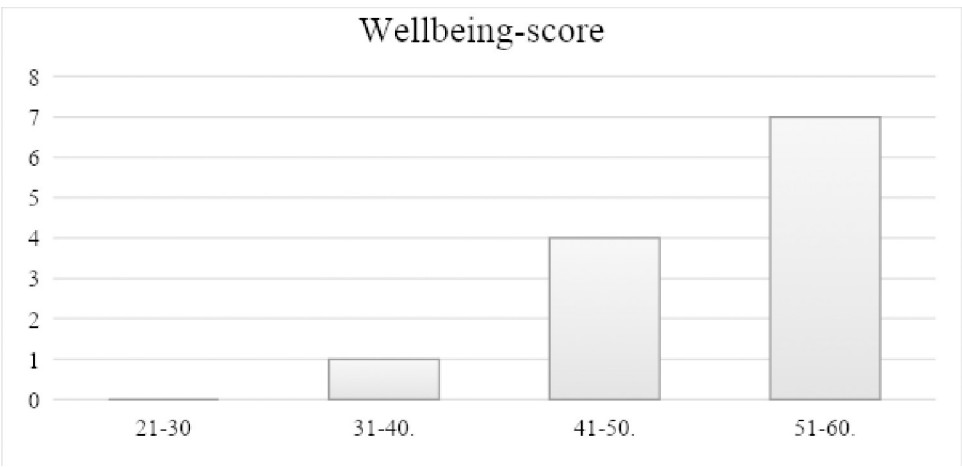

**Fig 1. Baseline WEMWBS wellbeing score of the sample at baseline.** N = 13, mean 51.6 SD: 6.6, x-axis: Wellbeing scale score points 14–70, stratified in deciles, y-axis: number of participants.

(54%) a mindfulness meditation session was 20–40 minutes long. Most participants had practiced mindfulness meditation for more than a year (59%) and had a total time on retreat of between 1 week and 1 month (38%). Among the participants, 46% mentioned stress reduction as their main motivation for practicing mindfulness meditation. 85% of the participants had already had what they considered a memorable experience in their mindfulness meditation, and 38,4% expected to have intense experiences during this study. A majority (54%) had not experienced profoundly altered states of mind at baseline, while 46% had experienced profoundly altered states of mind (not under the influence of drugs). Of the latter, 23% reported a lessened sense of self, lessened ownership of the body or feelings of unity. S1 Fig presents more detail on the participants' experience of mindfulness meditation at baseline.

### 3.2. Qualitative analysis

The qualitative analysis was based on the themes emerging from the participants' practice reports as illustrated in Table 2. The themes were structured into a mental and somatic theme

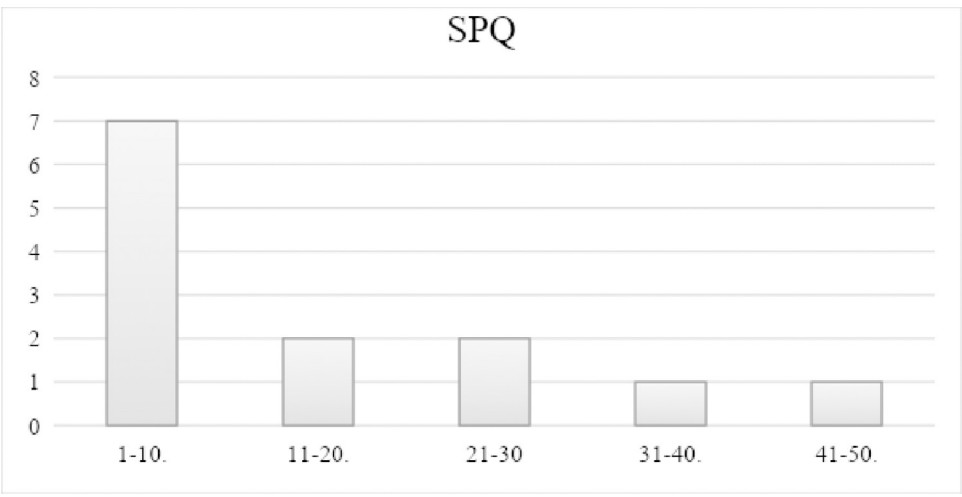

**Fig 2. Baseline SPQ score of the sample at baseline.** N = 13, mean 14.1, median 8, x-axis: SPQ points 0–74 stratified in deciles, y-axis: number of participants.

category. The mental and somatic theme categories were in turn organized into three groups of feeling tones: pleasant, unpleasant and neutral. Reports that were neither pleasant nor unpleasant were coded into the neutral theme. Further, those reports where the participant may have intended to indicate a pleasant or unpleasant tone to their experience but did not express this clearly enough were also coded into the neutral theme.

Below, the different themes are presented and contextualized with some example quotes. Each quote is identified with the report number in which they posted the quoted entry in chronological order (report 1, abbreviated as r1), the participant's identification number (e.g. number 3, abbreviated n3), their gender (F for female or M for male, O for other and P for prefer not to answer), and their age in years (e.g., age 34, abbreviated a34).

**3.2.1 Pleasant experiences in the mindfulness meditation practice.** The pleasant experience group had fewer references than both the negative and neutral experiences groups. In the pleasant experiences feeling tone, two theme categories emerged; mental and somatic. The mental theme category was divided into three further themes: affective, change in perception and cognitive characteristics. The affective characteristics group was the most common theme in the pleasant feeling tone group.

There were two features that distinguished the pleasant affective characteristics. One concerned emotional stability and an increased ability to accept and stay with difficulties. These features of the practice had a sustained positive influence on the participants' daily life. The other feature related to a sense of awe, wonder, gratitude and contentment, experiences that arose more commonly during the mindfulness meditation practice:

> *I eased into this awareness and then my tutor guided us to bring awareness to the heart and suddenly I felt a rush of gratitude flow throughout my whole body. Gratitude for being and for allowing me to be. (r2, n46, F, a27)*

A few participants experienced a change in the sense of self. They would describe how they became the object of focusing, the breath, awareness etc. This would usually be reported as something fairly strong, or deep experience:

> *I also had an experience of real expansiveness at a different sitting. My cushions face a window and I felt myself sort of merging into the blue sky. (r2, n20,F, a68)*

One participant expressed something that could best be expressed as magical thinking. During a body scan this participant expressed that she was aware of her brain and her telomeres in her mindfulness meditation. It was not clarified if this was described in a metaphorical way or if she actually felt she could "go through her telomeres". The pleasant somatic experiences concerned mainly relaxation and increased bodily awareness, as well as some increase in energy levels:

> *I became the body moving rhythmically, someone in the room coughed and I became the cough. My mind was still throughout this practice and I found the more I became the breath the more my body relaxed and settled into the posture. (r2, n46, F, a27)*

**3.2.2 Unpleasant experiences in the mindfulness meditation practice.** The unpleasant feeling tone group had more references than the pleasant, but less than the neutral feeling tone group. The unpleasant feeling tone group was divided into a somatic and a mental theme category, which in turn was divided into three themes. Some participants reported difficulties related to the mindfulness meditation practice. One participant reported switching completely

from mindfulness meditation to yoga due to difficulties, although the reason for the switch was not clearly expressed. Another participant found mindfulness meditation extremely difficult, although here as well there was no further explanation. Some participants experienced fear. Some experienced anxiety or panic, although they were not sure whether this was connected to mindfulness meditation. For some, feelings of intense sadness and trouble arose during the mindfulness meditation practice:

*I found myself really out of sorts, irritable, sad and troubled. I couldn't settle in meditation and had been sleeping badly. I felt as if I was feeling all the troubles of the world. (r5, n20, F, a68)*

Some participants experienced lessened control over their inner experience, which at times was difficult to handle:

*[I] have had a great deal of visual imaginary which happens internally, and which can, at times be disturbing. Often times I have no control over the imagery, it is not a hallucination, or thoughts but like watching a film in my head and I get the corresponding physical sensations. I have found this most definitely alienating in mindfulness groups. (r4, n5, O, a44)*

For some, mindfulness meditation had the effect of increased awareness of painful emotions. A couple of participants noted negative or repetitive thought patterns:

*I have noticed recurring thought and behavioral patterns within my meditations that do not serve me and with this awareness (r1, n46, F, a27)*

The most frequent somatic issues reported were difficulty in sleeping, tightness, pain and muscle contractions in the body, often correlating with a difficult mental experience:

*I find sitting extremely difficult at the moment as I am experiencing that when I do mindfulness it stops me sleeping. Also surges of energy and twitching, body jumping etc. I also seem to go into a trance quite easily. (r1, n5, O, a44)*

**3.2.3. Neutral experiences in the mindfulness meditation practice.**   The most common theme was that of experiences with a neutral feeling tone. The neutral theme was divided into three categories: somatic, mental and no notable experiences. The theme of no notable experiences was the most common one. This was the standard answer we suggested the practitioner could give to us had they not experienced anything they thought important to report. The mental category was divided into further subcategories: perceptual, cognitive and deep/intense experiences. Perceptual changes were the most frequent of these. In this category the need for an additional level of subcategory arose; changes in awareness. This included but was not limited to; expansions of awareness, changes in the awareness of the body, awareness of breathing and awareness of sensations. Aside from the category of awareness, deep experiences were reported. The most common reported cognitive change was noticing thought patterns. There were also some general changes in beliefs.

With regard to the changes in awareness some reported expansion of awareness:

*More spacious awareness (g3, r2, n52, F, a52)*

Some experienced changes in the perception of their bodies:

*I have been experimenting with different practices recently. In one guided meditation we were led to focus our energy on the third eye chakra and visualize a golden gate. During this meditation I felt intense pressure and then a blinking sensation in this area between the brows. I felt that the golden light filled up my body and entered through this portal. (g2, r1, n46, F, a27)*

Visual imagery was a common occurrence for the participants, sometimes with a significant impact on their perspective of life:

*As I was leading meditation, during one of the areas of silence I started to see white lights. Usually if I see colors they tend to begin with purple. (g3, r6, n25, M, a41)*

*I also appear to have lost all motivation in life, as I am so aware of death in my visual images that life seems fleeting and I find myself chuckling at how seriously other people seem to take it. (r4, n5, O, a44)*

Awareness of breath and mental sensations was easier during meditation:

*During the body scan my mind wandered and I became aware of the pattern my mind has of judging (especially myself) and being quite critical. (g2, w2, n46, F, a27)*

*I've come back around to creating more awareness of my breathing and noticing my sensations (mainly fear response) and allowing it or paying attention and expanding my awareness around this. (g1, r1, n30, F, a47)*

The most common reported cognitive change was noticing thought patterns. There were also some general changes in beliefs:

*I have noticed recurring thought and behavioral patterns. (g2, r1, n46, F, a27)*

*Changed beliefs about eating meat and drinking alcohol—changed habits—now sober and vegetarian. (g3, r4, n52, F, a54)*

The somatic experiences in the neutral arm were characterized by surges of energy, temperature changes and different types of feelings of pressure and twitching:

*Sometimes I also get surges of electricity running through my body. (r4, n5, O, a44)*

*During this meditation I felt intense pressure and then a blinking sensation in [the] area between the brows. (g2, r1, n46, F, a27)*

## 4. Discussion

### 4.1. Integration of results and previous evidence

This study investigated the nature and prevalence of intense experiences in mindfulness meditation practices as performed in the context of secular MBIs. The content of the participants' meditative experiences as put forward in their online reports showed a wide variety of mental and physical phenomena that could be organized into pleasant, unpleasant, and neutral experiences. The WEMWBS and HEPS scores show that our participants roughly reflect the general

population regarding wellbeing, meaning that these experiences may also appear in the general population when they practice mindfulness meditation.The qualitative analysis suggested that the practice of mindfulness meditation as taught in the context of MBIs might trigger the kind of emotionally intense experiences described in other mindfulness meditation research and case reports [38, 46–48].

There were many reports of pleasant experiences and for some people there were intense pleasant effects, e.g., a deep sense of calm or connectedness, that might be useful as a motivational aspect of practicing mindfulness. This is in line with previous research showing that mindfulness might be a useful tool for treating depression or anxiety [49]. However, pleasant experiences were the least frequently reported of the three feeling tones. It is also important to bear in mind that a very intense experience in mindfulness meditation, even if pleasant, might still be unsettling in that it might be unexpected or a cause for questions regarding its meaning and what it could lead to.

That difficult or unpleasant experiences were so frequent suggests that the mindfulness meditation practice even in the context of empirical MBIs might induce experiences in a wider emotional spectrum than previously thought, particularly considering that almost half of the participants had stress reduction as their main motivation. It has been noted that the practice of mindfulness might entail difficult experiences as well as the well-researched positive effects [49], and some scholars even suggest that the practice of mindfulness meditation can work as a stressor in vulnerable individuals [50]. However, these concerns and the nature of such difficult experiences have not been thoroughly investigated in the literature, partly due to previous studies lacking clear structures for detecting unwanted, unexpected, or adverse effects and events, resulting in a potential positivity bias, as noted by other scholars [28–31, 35]. In the unpleasant feeling tone group, affective negative emotions were common, and feelings of shame, sadness, fear and anxiety and for some even panics were reported. Many participants reported struggles, and some even changed to other practices due to their struggle, despite them being mindfulness teacher trainees.

In our qualitative analysis of the three feeling tones, the neutral tone had the richest data set. The largest theme numerically was that of no notable experiences, which was our suggested answer for participants if nothing of interest had happened since the last report two weeks earlier. Intense experiences do not seem to be happening often, otherwise this would most probably have been picked up in previous research.

Our participants gave many reports of intense experiences and some reports of what could be described as psychotic or psychotic-like experiences. Many participants experienced visual imagery of various sorts. The presence of lights, or other experiences of visual distortions are quite similar to subclinical psychotic-like phenomena [48]. Also, some participants reported changes in the perception of their bodies and sense of self, e.g., difficulties to discern the limit of the body relative to the surroundings. This is similar to bodily awareness distortions sometimes occurring during psychosis and depersonalisation [34, 36, 51]. Some participants also reported that their mindfulness meditation practice hindered their ability to sleep, which in itself may be a risk-factor for psychotic experiences [52].

On one hand, our results suggest that mindfulness meditation could potentially trigger intense experiences and in some cases psychotic-like phenomena like sensory distortions or changed self-perception. On the other hand, some evidence shows that mindfulness could be used to decrease symptoms described above, including psychosis [16, 47]. This description of the meditative process as an emotionally complex and sometimes conflicting practice is in line with the understanding of Lindahl who argues that the potential health-related benefits of mindfulness meditation exist but are merely a "narrow selection of possible effects that have been acknowledged within traditions both past and present" [29]. This also resonates with

many Buddhist literature sources describing mindfulness meditation as a process of both pleasant and unpleasant phenomena [53–55]. If we instead regard mindfulness practice as a process in which both pleasant, unpleasant and neutral experiences occur, that can be both intense and mild, this could potentially explain the sometimes-contradictory evidence and case reports of mindfulness meditation acting as both the cause as well as the remedy of some symptoms including psychotic episodes.

In some cases, similar experiences could for one participant be described as unpleasant, and for another pleasant or neutral. One example here was the visual imagery, which some found intriguing and beautiful, while others found them disturbing. Another explanation might be individual psychological or schizotypal differences: a person with a high SPQ score might attribute more metaphysical meaning or magical thinking to sensory experience than a low scoring SPQ person.

The mean SPQ score in our sample was slightly higher (14.1) than in a study of magical thinking and mindfulness conducted by Antonova et al. (11.62); however, the non-meditator group in that study scored 14.50. The study compared SPQ of 24 lay meditators (non-monks) and non-meditators in the UK. In our sample, the SPQ scores were not normally distributed, so the median score may be a better reference score [49]. If our sample's median score is taken into account, then both Antonova's and our meditators seem to have lower SPQ scores than non-meditators. Not only are we finding that meditators do not have higher SPQ scores than non-meditators, but they may even have lower SPQ scores. Therefore, in our small pilot study we do not find evidence that schizotypal personalities are more likely to meditate, so we cannot conclude that unusual experiences seen during mindfulness meditation are due to baseline schizotypal characteristics among meditators. This needs to be confirmed in future studies that compare SPQ scores of meditators with unusual experiences, with those among meditators without unusual experiences.

The fact that the participants were trained in different types of MBIs could potentially affect the results as the different MBIs may target different objectives. For example, MBSR has the explicit goal of general stress reduction while MBCT employs cognitive behavioural therapy to target symptoms of depression and negative thoughts. As MBIs are used for different aims, there is the possibility that people with different frames of reference interpret their experiences accordingly.

## 4.2. Strengths, limitations, and implications

As an initial approach to investigating a gap in the current literature, this study offers valuable insights for future research but also has important limitations. Results were based on 1–4 months collection of data, depending on time of entry of the participants; a longer follow-up may be needed for a more comprehensive study of the range of mindfulness meditative experiences. Also, due to the small sample and short follow up period we could not reliably assess whether experiences changed and evolved across the study period.

Another limitation was potentially, but not necessarily, the sample size. In the end, 13 participants were eligible to participate and contributed with data. A question that arose was if this makes for sufficient information power such as described by Malterud et al who propose a tradeoff between the size of a sample's information power, and the size of the sample. The research questions were clearly defined and the participants contributed with continuous data that varied within and between reports and participants, which makes for stronger information power. There is no intrinsic value to having a large sample size in conducting qualitative analysis, on the contrary; too large samples are potentially as bad as too small ones [43].

Due to the small size of the sample it is not possible to test for statistical significance comparing the groups. It would be important to investigate correlations between wellbeing, schizotypy, and meditative experiences.

Our results suggest that mindfulness or mindfulness meditation as taught and practiced in MBIs could potentially trigger a wider spectrum of experiences than previously thought [49]. The increasing knowledge of adverse effects and events linked to mindfulness practice, along with the known empirics of positive effects could be an important key to better understanding the full impact of mindfulness practice. This could also constitute a sound basis for further development of the strengths and limitations of MBIs. Our participants were not patients with clinical mental disorders, but rather fairly healthy individuals who were in the process of becoming mindfulness teachers (their wellbeing scores being higher, and their schizotypal median score being lower than that of the groups in Antonova's study [45]). These results suggest that even though mindfulness interventions have shown promising results, it might be good to consider preparing prospective mindfulness course participants for the fact that mindfulness-practice also might bring up difficulties as a part of the process. This is especially true for participants who may have particular sensitivities to the range of experiences.

It is important that future studies on the subject have clear structures to detect both positive and potential negative effects, and that researchers describe these findings in a clear way. More longitudinal studies with longer follow up to investigate how mindfulness meditation affects the participants over time, would therefore be of great value. Furthermore, complex interactions and temporal sequences may explain experiential phenomena arising from mindfulness meditation and how they impact health outcomes; there are descriptions of such patterns in the Buddhist literature, and Buddhist practices have greatly informed the development of mindfulness courses [21].

### 4.3. Concluding remarks

This study investigated the presence and nature of intense experiences caused by mindfulness practice through a qualitative analysis of online mindfulness meditation reports of mindfulness teachers. Interestingly, the unpleasant experiences were more frequently reported than pleasant experiences, possibly due to generally being more salient and therefore easier to remember than pleasant experiences. In all three feeling tones, intense experiences were reported, indicating that mindfulness practice affects the practitioner in more ways than merely relaxation and stress reduction, which was the main motivator and most common expectation for almost half of the sample at baseline. For some individuals these intense experiences may resemble psychotic-like experiences. Our results also indicate that some seemingly similar experiences could potentially have different impacts for different individuals, but also for the same individual at different times. Developing a more complex and nuanced understanding of the range of contemplative experiences seems necessary in context of the modern proliferation of mindfulness practice on the one hand, and the historical and contemporary reports of unusual or adverse experiences occasioned during the practice of mindfulness meditation, on the other.

### Supporting information

**S1 Fig. Hierarchical node tree.**
(DOCX)

### Author Contributions

**Conceptualization:** Erik Jönhagen, Tim Wood, Julieta Galante.

**Data curation:** Erik Jönhagen, Julieta Galante.

**Formal analysis:** Erik Jönhagen, Julieta Galante.

**Funding acquisition:** Julieta Galante.

**Investigation:** Erik Jönhagen, Tim Wood, Julieta Galante.

**Methodology:** Erik Jönhagen, Julieta Galante.

**Project administration:** Erik Jönhagen, Julieta Galante.

**Supervision:** Maria Niemi, Julieta Galante.

**Visualization:** Erik Jönhagen.

**Writing – original draft:** Erik Jönhagen, Julieta Galante.

**Writing – review & editing:** Erik Jönhagen, Tim Wood, Maria Niemi, Julieta Galante.

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
