## [Decision Letter · Decision Letter 0]

20 Oct 2023

PONE-D-23-15087Mindfulness Teacher Trainees' Experiences (MTTE) An investigation of intense experiences in mindfulness-based interventions and the risk of psychosis.PLOS ONE

Dear Dr. Galante,

Thank you for submitting your manuscript to PLOS ONE. After careful consideration, we feel that it has merit but does not fully meet PLOS ONE’s publication criteria as it currently stands. Therefore, we invite you to submit a revised version of the manuscript that addresses the points raised during the review process.

We look forward to receiving your revised manuscript.

Kind regards,

Eleni Petkari

Academic Editor

PLOS ONE

Journal Requirements:

4. Please upload a copy of Figure 3, to which you refer in your text on page 12. If the figure is no longer to be included as part of the submission please remove all reference to it within the text.

Reviewers' comments:

Reviewer's Responses to Questions

**Comments to the Author**

1. Is the manuscript technically sound, and do the data support the conclusions?

Reviewer #1: Partly

Reviewer #2: Partly

2. Has the statistical analysis been performed appropriately and rigorously? 

Reviewer #1: No

Reviewer #2: N/A

3. Have the authors made all data underlying the findings in their manuscript fully available?

Reviewer #1: Yes

Reviewer #2: Yes

4. Is the manuscript presented in an intelligible fashion and written in standard English?

Reviewer #1: Yes

Reviewer #2: Yes

5. Review Comments to the Author

Reviewer #1: I want to express my gratitude for your contribution to an area of mindfulness research that is often overlooked but of utmost importance. Your manuscript presents an insightful exploratory study that could serve as a valuable foundational resource for further research in this field. While the study is well-developed overall, there are specific areas where improvements are needed to enhance its rigour. My primary concern lies in the clarity of key concepts within the study, particularly the terms "mindfulness" and "meditation," which are used interchangeably without a clear rationale. Additionally, key terms like "intense experiences" require clarification. Moreover, it appears that a mixed-methods approach was employed, but the quantitative analysis has not been adequately introduced (even a brief mention, especially if descriptive statistics were used, would be beneficial). Furthermore, the Results and Discussion sections could benefit from more in-depth exploration and additional evidence to enrich these aspects of the paper. Below, I provide detailed suggestions for improvement:

Background

The introduction provides a thorough background on the topic, but it would benefit from improved clarity regarding some key concepts.

As mindfulness is a very broad term, it would be helpful if the author could provide a clearer definition of mindfulness for this study. While the authors listed several mindfulness practices in the Background section, it remains unclear which specific Mindfulness-Based Intervention (MBI) was used in this study.

A further distinction between mindfulness and meditation is needed as they are different concepts. It appears that the terms "meditation" and "mindfulness" are used interchangeably through the manuscript without providing a clear explanation.

A one-sentence definition/description would be helpful to clarify the term “intense experiences,” as this is a key measurement throughout your study. Is it the same as “psychotic disorders” or “psychotic-like experiences”? What are the constructs of “intense experiences”?

There is a typo after (14) on line 50; please review it.

You might find this article helpful for the background session:

• Phan-Le NT, Brennan L, Parker L (2022) The search for scientific meaning in mindfulness research: Insights from a scoping review. PLOS ONE 17(5): e0264924. https://doi.org/10.1371/journal.pone.0264924

as it provides classifications of mindfulness definitions and research domains.

Methods and Materials

The description of participant recruitment could benefit from additional information. Please clarify how many participants were recruited, what criteria were used to select them, and if there were any inclusion/exclusion criteria.

Why did the authors want to exclude participants with a Buddhist affiliation when selecting the participants? This needs to be explained. Is it because a Buddhist affiliation implies exposure to mindfulness?

Furthermore, did the authors take the “vulnerable group” into consideration when recruiting the sample? If so, it should also be stated in this session.

It might be helpful if the authors could also describe how the mindfulness training was conducted. According to section 3.1 (Description of the sample), it seems to refer merely to meditation. If that is the case, it would be helpful to explain why meditation was chosen as the main mindfulness practice for this study.

The exclusive criteria were clear, but the inclusive criteria were unclear. How did you ensure that the chosen participants went through an “intensive mindfulness training”?

In your Data Analysis section, while you have explained clearly how you performed the qualitative analysis, it is unclear how you analysed your quantitative data. Please provide a further description of your quantitative analysis for your survey data.

Results

Overall, the results section is informative and well-structured. However, in your Qualitative results, you focused mainly on meditation (evidently in your headings), but meditation is not the only mindfulness practice unless you explicitly chose meditation for this study (which is not stated in the Background or Methods sections). To make it more consistent and coherent, please either change the headings or explicitly state that meditation is the main mindfulness practice in the MBIs of this study.

The section on neutral experiences needs more evidence; currently, the chosen quotes do not clearly support your observations. Given that you mentioned that this theme has the richest data set, it is essential to provide stronger evidence.

I would expect more richness in this session. For example, is there any conflicting experience from some participants? Providing this is a longitudinal study, how the experiences change or evolve across the studied period?

Discussion

The discussion effectively integrates the study’s results with previous literature. One suggestion here is that the authors should discuss how different mindfulness training programs can potentially have an impact on the results, especially if the recruited participants are not following the same MBI program.

Furthermore, you should also include the discussion for the WEMWBS and the HEPS score.

Some references could be useful for this discussion, such as:

• Britton, Willoughby B., et al. "Defining and measuring meditation-related adverse effects in mindfulness-based programs." Clinical Psychological Science 9.6 (2021): 1185-1204.

• Newland, Pamela, and B. Ann Bettencourt. "Effectiveness of mindfulness-based art therapy for symptoms of anxiety, depression, and fatigue: A systematic review and meta-analysis." Complementary Therapies in Clinical Practice 41 (2020): 101246.

• Scheepers, RA, Emke, H, Epstein, RM, Lombarts, KMJMH. The impact of mindfulness-based interventions on doctors’ well-being and performance: A systematic review. Med Educ. 2020; 54: 138-149. https://doi.org/10.1111/medu.14020

• Zhang, Y., Chen, S., Wu, H. et al. Effect of Mindfulness on Psychological Distress and Well-being of Children and Adolescents: a Meta-analysis. Mindfulness 13, 285–300 (2022). https://doi.org/10.1007/s12671-021-01775-6

References

Some of your references are outdated. Please update your references to make them more relevant to today's context.

Reviewer #2: I am grateful to have the opportunity to review this intriguing research article titled "Mindfulness Teacher Trainees' Experiences: An Investigation of Intense Experiences in Mindfulness-Based Interventions and the Risk of Psychosis." The paper aims to explore the frequency and patterns of intense experiences within a cohort of mindfulness teachers with a minimum of one year of training. The 13 participants were recruited online between July and September 2019 and completed a questionnaire measuring wellbeing and schizotypal traits. Qualitative data was also collected to assess non-ordinary experiences. The study categorizes these experiences into two distinct categories: mental and somatic, further classifying them as pleasant, unpleasant, or neutral. According to the results, participants reported a variety of meditation experiences, with neutral experiences being the most prevalent, while pleasant and unpleasant experiences were less common.

Title

- Mindfulness Teacher Trainees' Experiences (MTTE): An Exploration of Intense Experiences in Mindfulness-Based Interventions and Their Relation to Psychosis Risk.

Introduction

1.1 Background

In this section, the authors provide some insights into the origins of mindfulness-based programs, including MBSR and MBCT. It is advisable to offer a more detailed distinction between these programs, elucidating their respective goals, target populations, and providing relevant references outlining their benefits and potential side effects. Furthermore, there is a typographical error on line 21 of page 9, where a hyphen is missing in "mindfulness-based cognitive therapy." It is essential to note that mindfulness-based interventions encompass both pleasant and unpleasant experiences, with an emphasis on well-being and the inclusion of practices and psychoeducation aimed at addressing challenging emotions. Authors should consider acknowledging this aspect in their description of MBIs.

1.2 Meditation and Psychotic Disorders / Mindfulness and Intense Experiences

In this section, the authors discuss various studies related to meditation and psychotic disorders, which is crucial for building the paper's narrative. Please be aware of the missing "T" in "The incidence" on line 50. While the authors mention unexpected and unwanted experiences resulting from mindfulness practice, ranging from intense emotions to altered perceptions and psychosis, the paper does not adequately link these experiences to mindfulness meditation. Reference 13 seems to relate to meditation practices but does not specify if it pertains to mindfulness practices or other categories of meditation. Additionally, the referenced study underscores that intense experiences are not uniquely tied to meditation but also influenced by factors such as fasting and sleep deprivation. I strongly recommend that the authors provide clearer information on this topic.

The authors should consider citing one of the most renowned works on the adverse effects of meditation practices:

- Recommended study and citation: Goldberg SB, Lam SU, Britton WB, Davidson RJ. Prevalence of meditation-related adverse effects in a population-based sample in the United States. Psychother Res. 2022 Mar;32(3):291-305. doi: 10.1080/10503307.2021.1933646. Epub 2021 Jun 2. PMID: 34074221; PMCID: PMC8636531.

Updating and reviewing the references on this topic is of paramount importance.

On line 58, please correct the citation format to something like:

- (8, 9, 10, 11, 34)

1.4 Aim

The authors could replace "described above" with a more explicit description of the study's objectives to enhance clarity for the reader.

Discussion/Conclusion

In this section, the authors outline the study's aim, which is to investigate the nature and prevalence of intense experiences during meditation or mindfulness practices within the context of secular MBIs. It would be beneficial if the authors provided a clear definition of "intense experiences" and offered additional clarity regarding the participants' perceptions of the intensity of their reported meditation experiences. Exploring why participants tend to report more neutral and unpleasant experiences than pleasant ones and considering whether this may be linked to an evolutionary predisposition towards being more aware of unpleasant experiences would be an interesting addition. Readers would appreciate the authors' interpretations of this result.

In line 471-472, the authors suggest that unpleasant experiences were the most commonly reported, which contradicts the information provided in the results section, where neutral experiences were identified as the most frequently reported by the sample in the study.

While the authors present important data on the affective nature of meditation experiences in mindfulness teachers, a more detailed description of the characteristics of the practices would be valuable, including their duration and type (focused attention or open monitoring).

The title and some conclusions drawn by the authors do not appear to completely align with the data presented, especially concerning the risk of psychotic episodes or psychotic experiences after mindfulness practices. The paper seems to take a direction that does not necessarily correspond to the study's results.

6. PLOS authors have the option to publish the peer review history of their article (what does this mean?). If published, this will include your full peer review and any attached files.

Reviewer #1: **Yes: **Nhat Tram Phan-Le

Reviewer #2: No

---

## [Author Response · Author response to Decision Letter 0]

17 Feb 2024

We wish to thank the reviewers for their thoughtful comments. Please find our replies below.

Reviewer #1 

I want to express my gratitude for your contribution to an area of mindfulness research that is often overlooked but of utmost importance. Your manuscript presents an insightful exploratory study that could serve as a valuable foundational resource for further research in this field. While the study is well-developed overall, there are specific areas where improvements are needed to enhance its rigour. My primary concern lies in the clarity of key concepts within the study, particularly the terms "mindfulness" and "meditation," which are used interchangeably without a clear rationale. Additionally, key terms like "intense experiences" require clarification. Moreover, it appears that a mixed-methods approach was employed, but the quantitative analysis has not been adequately introduced (even a brief mention, especially if descriptive statistics were used, would be beneficial). Furthermore, the Results and Discussion sections could benefit from more in-depth exploration and additional evidence to enrich these aspects of the paper. Below, I provide detailed suggestions for improvement:

Background

The introduction provides a thorough background on the topic, but it would benefit from improved clarity regarding some key concepts.

As mindfulness is a very broad term, it would be helpful if the author could provide a clearer definition of mindfulness for this study.

REPLY: You are quite correct with the need for a clearer definition. We have clarified this in the paper: 

”In the scientific literature, the term “mindfulness” is used to describe quite different types of processes and it is subject to a variety of definitions (4, 9). There is also an ongoing discussion regarding the scientific implications on the varied definitions of mindfulness across domains (10). In this paper we will use Kabat Zinn’s definition as it is the most typical definition used in MBIs. Mindfulness is defined as “the non-judgmental acceptance and investigation of present experience, including body sensations, internal mental states, thoughts, emotions, impulses and memories, in order to reduce suffering or distress and to increase well-being” (4)”. (line 25 in unmarked version). 

While the authors listed several mindfulness practices in the Background section, it remains unclear which specific Mindfulness-Based Intervention (MBI) was used in this study.

REPLY: Yes, we have now added the exact inclusion criteria which states that participants needed to be trained in an MBI program including but not limited to MBSR and MBCT (line 130). 

A further distinction between mindfulness and meditation is needed as they are different concepts. It appears that the terms "meditation" and "mindfulness" are used interchangeably through the manuscript without providing a clear explanation.

REPLY: Yes, we have now replaced all meditation with “mindfulness meditation” which is also now more clearly defined as described above.

A one-sentence definition/description would be helpful to clarify the term “intense experiences,” as this is a key measurement throughout your study. Is it the same as “psychotic disorders” or “psychotic-like experiences”? What are the constructs of “intense experiences”?

REPLY: This is a good point. Many scholars use the term without clearly defining its characteristics. We have added our working definition of intense experiences as a strong sensate or psychological experience during mindfulness meditation that clearly differs from normal expectations of stress reduction (line 49). 

There is a typo after (14) on line 50; please review it.

REPLY: Thank you for spotting this, it has been corrected. 

You might find this article helpful for the background session:

• Phan-Le NT, Brennan L, Parker L (2022) The search for scientific meaning in mindfulness research: Insights from a scoping review. PLOS ONE 17(5): e0264924.https://doi.org/10.1371/journal.pone.0264924

as it provides classifications of mindfulness definitions and research domains.

REPLY: Thank you for this, we have included it in the paper. 

Methods and Materials

The description of participant recruitment could benefit from additional information. Please clarify how many participants were recruited, what criteria were used to select them, and if there were any inclusion/exclusion criteria.

REPLY: The inclusion criteria and number of participants are now fully spelled out in the beginning of the methods section:

” This was an online prospective longitudinal study following 13 secular mindfulness teacher trainees. We recruited online between July and September 2019 through mindfulness teacher networks in English-speaking countries around the world. Inclusion criteria were as follows: 1. Secular mindfulness-based programme (including but not limited to MBSR and MBCT) teacher-in-training, or a teacher who finished their training within the past year; 2. No regular (i.e. at least 20 minutes daily or near-daily) meditation practice until the year prior to starting teacher training; 3. No or little exposure to Buddhist teachings (e.g. have not attended Buddhist retreats). The reason we excluded participants with a Buddhist affiliation is because we wanted to understand the occurrence and appraisal of intense mindfulness meditation experiences among modern secular practitioners. Having a Buddhist worldview could have predisposed the participants to have specific spiritual experiences, or provided a specific spiritual framework to interpret any intense experience; 4. No other previous meditation teacher training (including Yoga teacher training); 5. Residing in the United States, United Kingdom Ireland, South Africa, Canada, Australia or New Zealand; 6. Over 18 years old.”

Why did the authors want to exclude participants with a Buddhist affiliation when selecting the participants? This needs to be explained. Is it because a Buddhist affiliation implies exposure to mindfulness?

REPLY: The reason we excluded participants with a Buddhist affiliation is because we wanted to understand the occurrence and appraisal of intense mindfulness meditation experiences among modern secular practitioners. Having a Buddhist worldview could have predisposed the participants to have specific spiritual experiences, or provided a specific spiritual framework to interpret any intense experience. We have now added this clarification (line 136).

Furthermore, did the authors take the “vulnerable group” into consideration when recruiting the sample? If so, it should also be stated in this session. 

REPLY: Not when we recruited the sample. However, we offered support by medical professionals in the event that reporting these experiences would evoke difficulties.

It might be helpful if the authors could also describe how the mindfulness training was conducted. According to section 3.1 (Description of the sample), it seems to refer merely to meditation. If that is the case, it would be helpful to explain why meditation was chosen as the main mindfulness practice for this study.

REPLY: We would like to clarify that we did not conduct any mindfulness training for this study. We recruited participants who were training to be MBI teachers. We hope that the addition of the selection criteria clarifies this.

The exclusive criteria were clear, but the inclusive criteria were unclear. How did you ensure that the chosen participants went through an “intensive mindfulness training”?

REPLY: Now the inclusion criteria is more clearly depicted in the method section.The aim was not to go through intensive mindfulness training. Rather, the idea was to investigate whether intense experiences might arise in MBIs. 

In your Data Analysis section, while you have explained clearly how you performed the qualitative analysis, it is unclear how you analysed your quantitative data. Please provide a further description of your quantitative analysis for your survey data.

REPLY: Yes, we have now clarified this in the data analysis section (line 164). 

Results

Overall, the results section is informative and well-structured. However, in your Qualitative results, you focused mainly on meditation (evidently in your headings), but meditation is not the only mindfulness practice unless you explicitly chose meditation for this study (which is not stated in the Background or Methods sections). To make it more consistent and coherent, please either change the headings or explicitly state that meditation is the main mindfulness practice in the MBIs of this study.

REPLY: We have now clarified this throughout the paper using mindfulness meditation and not meditation.

The section on neutral experiences needs more evidence; currently, the chosen quotes do not clearly support your observations. Given that you mentioned that this theme has the richest data set, it is essential to provide stronger evidence.

REPLY: This is a good point. We have now added more quotes in the neutral part.

I would expect more richness in this session. For example, is there any conflicting experience from some participants? Providing this is a longitudinal study, how the experiences change or evolve across the studied period?

REPLY: We agree on this part and have added a paragraph in the limitations section addressing your concerns (line 537). We also are planning further research that will look into this precisely.

Discussion

The discussion effectively integrates the study’s results with previous literature. One suggestion here is that the authors should discuss how different mindfulness training programs can potentially have an impact on the results, especially if the recruited participants are not following the same MBI program.

REPLY: We now address this as a limitation (line 525).

Furthermore, you should also include the discussion for the WEMWBS and the HEPS score.

REPLy: Yes, we have added a short part about this. Section 4.1, line 442: “The WEMWBS and HEPS scores show that our participants roughly reflect the general population regarding mental wellbeing, meaning that these experiences may also appear in the general population when they practice mindfulness meditation.”

Some references could be useful for this discussion, such as:

• Britton, Willoughby B., et al. "Defining and measuring meditation-related adverse effects in mindfulness-based programs." Clinical Psychological Science 9.6 (2021): 1185-1204.

• Newland, Pamela, and B. Ann Bettencourt. "Effectiveness of mindfulness-based art therapy for symptoms of anxiety, depression, and fatigue: A systematic review and meta-analysis." Complementary Therapies in Clinical Practice 41 (2020): 101246.

• Scheepers, RA, Emke, H, Epstein, RM, Lombarts, KMJMH. The impact of mindfulness-based interventions on doctors’ well-being and performance: A systematic review. Med Educ. 2020; 54: 138-149. https://doi.org/10.1111/medu.14020

• Zhang, Y., Chen, S., Wu, H. et al. Effect of Mindfulness on Psychological Distress and Well-being of Children and Adolescents: a Meta-analysis. Mindfulness 13, 285–300 (2022). https://doi.org/10.1007/s12671-021-01775-6

REPLY: Thank you for these suggestions. We have found that the Britton paper very much aligns with this paper and it has been added, though in the background section 1.3.

References

Some of your references are outdated. Please update your references to make them more relevant to today's context.

REPLY: We have now done this.

Reviewer #2

I am grateful to have the opportunity to review this intriguing research article titled "Mindfulness Teacher Trainees' Experiences: An Investigation of Intense Experiences in Mindfulness-Based Interventions and the Risk of Psychosis." The paper aims to explore the frequency and patterns of intense experiences within a cohort of mindfulness teachers with a minimum of one year of training. The 13 participants were recruited online between July and September 2019 and completed a questionnaire measuring wellbeing and schizotypal traits. Qualitative data was also collected to assess non-ordinary experiences. The study categorizes these experiences into two distinct categories: mental and somatic, further classifying them as pleasant, unpleasant, or neutral. According to the results, participants reported a variety of meditation experiences, with neutral experiences being the most prevalent, while pleasant and unpleasant experiences were less common.

Title

- Mindfulness Teacher Trainees' Experiences (MTTE): An Exploration of Intense Experiences in Mindfulness-Based Interventions and Their Relation to Psychosis Risk.

Introduction

1.1 Background

In this section, the authors provide some insights into the origins of mindfulness-based programs, including MBSR and MBCT. It is advisable to offer a more detailed distinction between these programs, elucidating their respective goals, target populations, and providing relevant references outlining their benefits and potential side effects. Furthermore, there is a typographical error on line 21 of page 9, where a hyphen is missing in "mindfulness-based cognitive therapy." It is essential to note that mindfulness-based interventions encompass both pleasant and unpleasant experiences, with an emphasis on well-being and the inclusion of practices and psychoeducation aimed at addressing challenging emotions. Authors should consider acknowledging this aspect in their description of MBIs.

REPLY: Thank you for your suggestion. We have added this in the background section, line 32.

1.2 Meditation and Psychotic Disorders / Mindfulness and Intense Experiences

In this section, the authors discuss various studies related to meditation and psychotic disorders, which is crucial for building the paper's narrative. Please be aware of the missing "T" in "The incidence" on line 50. While the authors mention unexpected and unwanted experiences resulting from mindfulness practice, ranging from intense emotions to altered perceptions and psychosis, the paper does not adequately link these experiences to mindfulness meditation. Reference 13 seems to relate to meditation practices but does not specify if it pertains to mindfulness practices or other categories of meditation. Additionally, the referenced study underscores that intense experiences are not uniquely tied to meditation but also influenced by factors such as fasting and sleep deprivation. I strongly recommend that the authors provide clearer information on this topic.

REPLY: Thank you for your suggestion, we have expanded this section.

The authors should consider citing one of the most renowned works on the adverse effects of meditation practices:

- Recommended study and citation: Goldberg SB, Lam SU, Britton WB, Davidson RJ. Prevalence of meditation-related adverse effects in a population-based sample in the United States. Psychother Res. 2022 Mar;32(3):291-305. doi: 10.1080/10503307.2021.1933646. Epub 2021 Jun 2. PMID: 34074221; PMCID: PMC8636531.

Updating and reviewing the references on this topic is of paramount importance.

REPLY: Thank you for this suggestion. It has been added to the section 1.3.

On line 58, please correct the citation format to something like:

- (8, 9, 10, 11, 34)

REPLY: This has been corrected.

1.4 Aim

The authors could replace "described above" with a more explicit description of the study's objectives to enhance clarity for the reader.

REPLY: Thank you, we have clarified this.

Discussion/Conclusion

In this section, the authors outline the study's aim, which is to investigate the nature and prevalence of intense experiences during meditation or mindfulness practices within the context of secular MBIs. It would be beneficial if the authors provided a clear definition of "intense experiences" and offered additional clarity regarding the participants' perceptions of the intensity of their reported meditation experiences. Exploring why participants tend to report more neutral and unpleasant experiences than pleasant ones and considering whether this may be linked to an evolutionary predisposition towards being more aware of unpleasant experiences would be an interesting addition. Readers would appreciate the authors' interpretations of this result.

REPLY: The introduction now provides a definition section with a definition of intense experiences (line 49). We have also added a comment on the possible reason for the higher frequency of reported unpleasant experiences. 

In line 471-472, the authors suggest that unpleasant experiences were the most commonly reported, which contradicts the information provided in the results section, where neutral experiences were identified as the most frequently reported by the sample in the study.

REPLY: Thank you, we have corrected this.

While the authors present important data on the affective nature of meditation experiences in mindfulness teachers, a more detailed description of the characteristics of the practices would be valuable, including their duration and type (focused attention or open monitoring).

REPLY: We have added some information on this in the background section. 1.1 (line 35). 

The title and some conclusions drawn by the authors do not appear to completely align with the data presented, especially concerning the risk of psychotic episodes or psychotic experiences after mindfulness practices. The paper seems to take a direction that does not necessarily correspond to the study's results.

REPLY: Thank you, we have changed the title and added a few lines (489 and 583) to integrate the paper better.

---

## [Decision Letter · Decision Letter 1]

19 Mar 2024

Mindfulness Teacher Trainees' Experiences (MTTE) An investigation of intense experiences in mindfulness-based interventions

PONE-D-23-15087R1

Dear Dr. Galante,

We’re pleased to inform you that your manuscript has been judged scientifically suitable for publication and will be formally accepted for publication once it meets all outstanding technical requirements.

Kind regards,

Eleni Petkari

Academic Editor

PLOS ONE

Additional Editor Comments (optional):

Reviewers' comments:

Reviewer's Responses to Questions

**Comments to the Author**

1. If the authors have adequately addressed your comments raised in a previous round of review and you feel that this manuscript is now acceptable for publication, you may indicate that here to bypass the “Comments to the Author” section, enter your conflict of interest statement in the “Confidential to Editor” section, and submit your "Accept" recommendation.

Reviewer #1: All comments have been addressed

2. Is the manuscript technically sound, and do the data support the conclusions?

Reviewer #1: Yes

3. Has the statistical analysis been performed appropriately and rigorously? 

Reviewer #1: Yes

4. Have the authors made all data underlying the findings in their manuscript fully available?

Reviewer #1: Yes

5. Is the manuscript presented in an intelligible fashion and written in standard English?

Reviewer #1: Yes

6. Review Comments to the Author

Reviewer #1: I express my gratitude to the authors for considering and incorporating feedback from this reviewer when suitable. The revised manuscript has adequately responded to all remarks. This research area of mindfulness is essential and will undoubtedly serve as a valuable resource for scholars to address the concerns towards intense experience of mindfulness. .

7. PLOS authors have the option to publish the peer review history of their article (what does this mean?). If published, this will include your full peer review and any attached files.

Reviewer #1: **Yes: **Nhat Tram Phan-Le

---

## [Editor Report · Acceptance letter]

25 Mar 2024

PONE-D-23-15087R1 

PLOS ONE

Dear Dr. Galante, 

I'm pleased to inform you that your manuscript has been deemed suitable for publication in PLOS ONE. Congratulations! Your manuscript is now being handed over to our production team.

Kind regards, 

on behalf of

Dr. Eleni Petkari 

Academic Editor

PLOS ONE